# Design and Synthesis of a New Mannitol Stearate Ester-Based Aluminum Alkoxide as a Novel Tri-Functional Additive for Poly(Vinyl Chloride) and Its Synergistic Effect with Zinc Stearate

**DOI:** 10.3390/polym11061031

**Published:** 2019-06-11

**Authors:** Wenyuan Han, Manqi Zhang, Degang Li, Tianbao Dong, Bing Ai, Jianping Dou, Hongqi Sun

**Affiliations:** 1School of Chemistry and Chemical Engineering, Shandong University of Technology, Zibo 255000, China; 18753375118@163.com (W.H.); 17853321663@163.com (M.Z.); 15864046907@163.com (T.D.); hgxyaibing@163.com (B.A.); 2Zibo Environmental Monitoring Station, Zibo 255000, China; doujianping@zb.shandong.cn; 3School of Engineering, Edith Cowan University, 270 Joondalup Drive, Joondalup, WA 6027, Australia

**Keywords:** poly(vinyl chloride), tri-functional additive, mannitol stearate ester-based aluminum alkoxides, synergistic effect

## Abstract

Thermal stabilizers, lubricant, and plasticizers are three crucial additives for processing poly(vinyl chloride) (PVC). In this study, a new mannitol stearate ester-based aluminum alkoxide (MSE-Al) was designed and synthesized as a novel additive for PVC. The thermal stability and processing performance of PVC stabilized by MSE-Al were evaluated by the Congo red test, conductivity measurement, thermal aging test, ultravioletevisible (UV–Vis) spectroscopy test, and torque rheometer test. Results showed that the addition of MSE-Al could not only markedly improve the long-term thermal stability of PVC, but also greatly accelerate the plasticizing and decrease the balance torque, which demonstrated that MSE-Al possessed a lubricating property. Thus, MSE-Al was demonstrated to be able to provide tri-functional additive roles, e.g., thermal stabilizer, plasticizer, and lubricant. The test results for the thermal stability of PVC indicated that the initial whiteness of PVC stabilized by MSE-Al was not good enough, thus the synergistic effect of MSE-Al with zinc stearates (ZnSt_2_) on the thermal stability of PVC was also investigated. The results showed that there is an appreciable synergistic effect between MSE-Al and ZnSt_2_. The thermal stabilization mechanism and synergism effect of MSE-Al with ZnSt_2_ are then discussed.

## 1. Introduction

Poly(vinyl chloride) (PVC) is formed by the polymerization of a vinyl chloride monomer. PVC has many attractive characteristics, for example, corrosion resistance, wear resistance, flame retardancy and insulation, and therefore has been widely used in industries, agriculture, construction, electric power, and so on. PVC has become one of the most widely used plastics in the world because of its outstanding properties [1,2,3]. Due to defects in the chain structure, there is a small amount of unstable chlorine in PVC. When PVC is exposed to heat, ultraviolet radiation, etc., unstable chlorine atoms are released in the form of hydrogen chloride. Unsaturated C–C bonds will appear in the molecular chains of PVC. The unsaturated C–C bonds will lead to newly emerged unstable chlorine atoms, for instance, allyl chloride. Thermal degradation would normally occur when PVC is heated to more than 100 °C, while the processing temperature of PVC is 180 °C. Thus, thermal stabilizers must be added to improve the thermal stability of PVC during processing. Furthermore, being a strong Lewis acid, hydrogen chloride produced from the degradation of PVC catalyzes the further degradation of PVC. This leads to the formation of a “zipper” dehydrochlorination reaction, resulting in a change in the color of PVC (from white to brown, and finally to black), which further changes the properties of the PVC [4,5].

In general, PVC thermal stabilizers should have at least two functions: (1) replacing or passivating labile chlorine atoms in the PVC chain, such as allyl chloride atoms, preventing the formation of conjugated double bonds; and (2) absorbing or neutralizing HCl produced during the degradation of PVC to inhibit the autocatalytic dehydrochlorination reaction [6,7,8]. At present, common thermal stabilizers of PVC include lead salts [9], organotin [6], rare earth [10], calcium and zinc soap salts (especially the calcium and zinc stearates) [11]. The lead salts have an excellent long-term thermal stability, simple production process and low price. However, due to the toxicity on humans and the environment, their use has been restricted. The organotin thermal stabilizers are effective, but they are often used in high-grade PVC products because of the complex production processes and high production costs, which reduce their competitiveness with other thermal stabilizers. Moreover, certain organotins based on mercaptides also generate unpleasant odors [6]. The rare earth thermal stabilizers are similar to organotin, but more expensive than the others, and the cost hinders the widespread application of rare earth thermal stabilizers. Ca/Zn soap salts are non-toxic, environmentally friendly thermal stabilizers. Calcium stearate (CaSt_2_) and zinc stearate (ZnSt_2_) stabilizers contain fatty acid root, which gives these thermal stabilizers good lubrication properties, and they are relatively easy to process [12,13]. Moreover, ZnSt_2_ has an ability to replace the active chloride atom and then significantly increase the initial color of PVC. Thus PVC products with Ca/Zn thermal stabilizers have a good initial color and smooth surface characteristics. However, the reaction of ZnSt_2_ to replace the active chloride atom and to absorb HCl will produce zinc chloride (ZnCl_2_). ZnCl_2_, as a Lewis acid that can play a catalytic role in the degradation of PVC, would further accelerate degradation of PVC. When the accumulation of ZnCl_2_ reaches a certain amount, the degradation rate of PVC will increase suddenly, making the PVC product quickly turn black. This phenomenon is called “zinc burning” [14,15]. In order to avoid this phenomenon, ZnSt_2_ is often used together with calcium stearate (CaSt_2_). After ZnSt_2_ is reacted with HCl to produce ZnCl_2_, it can continue to react with CaSt_2_ to form CaCl_2_ and ZnSt_2_. CaCl_2_ does not catalyze the degradation of PVC, and therefore it inhibits “zinc burning” over a period of time [13].

As an important organic auxiliary thermal stabilizer, some polyols, for example, pentaerythritol, sorbitol, trimethylolpropane, etc., are often used in combination with CaSt_2_/ZnSt_2_ stabilizers to improve the long-term thermal stability of PVC [16]. Polyols have a large number of hydroxyl groups that can form a stable chelate with ZnCl_2_ to inhibit its further catalytic degradation of PVC. In previous studies, we reported the synthesis of lanthanum-pentaerythritol alkoxides [17], and aluminum-pentaerythritol alkoxides [18], and their improved performances for the thermal stability of PVC. However, most polyol-based metal alkoxides have high melting points, which make them less dispersible in PVC. In addition, polyols have a certain degree of water solubility, resulting in poor miscibility with PVC and poor PVC plasticization. The presence of these drawbacks limits the widespread use of polyol-based metal alkoxides.

In this study, mannitol stearate ester (MSE) was synthesized through a direct esterifying reaction between mannitol and stearic acid, then the mannitol stearate ester-based aluminum alkoxide (MSE-Al) was prepared through the alcohol exchange reaction between MSE and aluminum isopropoxide [19,20]. Compared to pentaerythritol, mannitol has a lower melting point of 166 °C and possesses a greater number of hydroxyl groups per unit mass. The design idea was just to improve the lubricity and reduce the melting point of MSE-Al to obtain a better compatibility with PVC by introducing a stearate functional group. The results of the thermal stability test further indicated that MSE-Al could provide tri-functional additive roles, namely thermal stabilizer, plasticizer and lubricant.

## 2. Experimental

### 2.1. Materials

PVC resin (average polymerization degree of 1005) was purchased from China Petrochemical Qilu Co. Ltd., Zibo, China. Some additives, such as lead salts stabilizers (the mixture of tribasic lead sulfate, dibasic lead phosphite, polyethylene wax, and assistant agents, PbO content: 30 ± 2%), ZnSt_2_ (98%), CaCO_3_ (light calcium carbonate, 1800 mesh), chlorinated polyethylene (CPE, Chlorine content: 35 ± 2%), dioctyl phthalate (DOP, 99%), TiO_2_ (anatase titanium dioxide, 99%), and acrylics copolymer (ACR, 99%), were all of industrial grade and kindly supplied by Shandong Huike Additives Co., Zibo, China. Stearic acid (≥98%), D-mannitol (≥99.0%), cyclohexane (≥99.0%), aluminum isopropoxide (≥98%) and other chemical agents were all of analytical grade and supplied by Shanghai McLean biochemical technology Co., Ltd., Shanghai, China.

### 2.2. Preparation of MSE and MSE-Al

MSE was prepared through a direct esterifying reaction between D-mannitol and stearic acid. Stearic acid (0.1 mol, 28.4 g) and D-Mannitol (0.1 mol, 18.2 g) were mixed together with 0.3 g p-toluenesulfonic acid (used as a catalyst) in a mixer set. Then the mixture and 10 mL of cyclohexane (used as water-carrying agent) were put into a three-necked round bottom flask equipped with a water separator and an electric stirrer. The mixture was heated at 170 °C for 4 h under a nitrogen atmosphere. After evaporating cyclohexane, the MSE was removed and ground into powders. Then they were put into a vacuum oven and dried for 12 h at 45 °C. Scheme 1 illustrates the synthesis pathway of MSE.

To determine the yield of the esterification reaction for MSE, the volume of water produced after the completion of the esterification reaction was measured. By comparing the actual water volume (*V*_a_) and the theoretical water volume (*V*_t_), the yield of MSE could be calculated using Equation (1). The yield of MSE from Equation (1) was 96% ± 1%.
(1)$$\mathrm{The}\;\mathrm{yield}\;\mathrm{of}\;\mathrm{MSE}\;=\;\frac{V_{a}}{V_{t}}\cdot 100\%$$

MSE-Al was prepared by the alcohol exchange reaction between MSE and aluminum isopropoxide. MSE (20 mmol, 8.97 g) and aluminum isopropoxide (10 mmol, 2.04 g) were dissolved in 100 mL absolute ethanol in a 250 mL three-necked round bottom flask equipped with a magnetic stirrer and a condenser-Allihn type. The mixed solution was heated to 140 °C with stirring and refluxing under a nitrogen atmosphere for 3 h, and then MSE-Al was obtained by evaporating the solvent. The white solid was put into a vacuum drying oven and fully dried for 8 h at 45 °C, then grounded into powder for later use. Scheme 2 illustrates the synthesis pathway of MSE-Al.

As aluminum isopropoxide and the produced isopropanol were soluble in absolute ethanol, while MSE-Al was insoluble in absolute ethanol at room temperature, the crude product of MSE-Al was smashed and put into absolute ethanol. The mixture was filtered after stirring for 1 h. The filtrate components were ethanol, isopropanol and unreacted aluminum isopropoxide. After the liquid was evaporated at 90 °C, the residual solid (aluminum isopropoxide) was measured by inductively coupled plasma mass spectrometry (ICP-MS). By comparing the molar amount of aluminum in the unreacted aluminum isopropoxide (*M*_u_/mmol) and the total molar amount of aluminum (10 mmol) in the reactant, the yield of MSE-Al could be calculated through Equation (2). The yield of MSE-Al from Equation (2) was 92% ± 2%.
(2)$$\mathrm{The}\;\mathrm{yield}\;\mathrm{of}\;\mathrm{MSE}-\mathrm{Al}\;=\;\frac{10-M_{u}}{10}\cdot 100\%$$

### 2.3. Characterization of MSE and MSE-Al

Fourier-transform infrared (FT-IR) spectra of MSE and MSE-Al were obtained on a Nicolet 5700 spectrometer (ThermoElectron, Madison, WI, US) by the KBr disc method. The spectrum range was 4000–400 cm^−1^ with 120 scans at a resolution of 4 cm^−1^. Thermal gravimetric analysis (TGA) of MSE-Al was carried out by a Netzsch STA 449C (Selb, Germany) at a heating rate of 10 °C/min from 25 to 800 °C under nitrogen flow (20 mL·min^−1^).

### 2.4. Preparation of PVC Samples

The PVC master batch consisted of 100.0 g of PVC, 20.0 g of CaCO_3_, 9.0 g of chlorinated polyethylene (CPE), 4.0 g of TiO_2_, 2.5 g of dioctylphthalate (DOP), 2.0 g of acrylics copolymer (ACR) and 1.6 g of stearic acid (HSt), which were mixed together with 4.0 g of thermal stabilizers in a mixer set. Then the mixture was rolled using an open twin roller (XH-401, Dongguan Xihua Testing Machine Co., Ltd., Dongguan, China.) for 5 min at 180 °C. The thickness of pressed PVC sheets was about 1.0 ± 0.1 mm.

### 2.5. Thermal Stability Test of MSE-Al

#### 2.5.1. Congo Red Test

According to the standard of ISO 182-1-1990 [21], approximately 2 g of PVC sample was cut into 2 mm squares and put into a test tube in which the wet Congo red paper was placed about 2.0 cm over the sample. The test tube bottom was immersed into an oil bath at 180 °C, during which the oil was ensured to be higher than PVC samples. The time that Congo red test paper began to turn blue was defined as the thermal stability time (Δ*T*).

#### 2.5.2. Conductivity Measurement

PVC (2 g) squares of 2.0 mm × 2.0 mm were put into the reaction vessel heated at 180 °C. Nitrogen (about 7 L/h) was introduced into the reaction vessel and blown out from the outlet, then was passed into 60 mL of deionized water. The HCl gas formed from the thermal degradation of PVC would be carried by the nitrogen gas and absorbed by the deionized water, resulting in changes in the conductivity of the water with respect to time [6]. The change in conductivity reflected the degradation rate of PVC. The conductivity meter used to measure the conductivity of solution was made by Shanghai INESA Scientific Instrument Company Limited, Shanghai, China (DDS-307).

#### 2.5.3. Thermal Aging Test

According to the ISO 305-1990 (E) standard [22], PVC samples were cut into sheets of 15 mm × 15 mm size, which were put into a thermal aging test box and heated to 180 ± 2 °C. The PVC sheets were taken out after every 10 min. The color changes of PVC samples were observed and compared, which reflected thermal stability of the thermal stabilizers on PVC.

#### 2.5.4. UV–Vis Spectroscopy Test

With the HCl release during the thermal degradation of PVC, a conjugated double bond structure is simultaneously produced. The length and concentration of the conjugated double bond in PVC can be characterized by the wavelength and the height of the absorption peak in the UV–Vis spectrum, which was measured by the UV–Visible spectrometer (UV-2450PC, Shimadzu Scientific Instruments, Kyoto, Japan) with the slit width set at 2 nm over the wavelength in the range of 270–400 nm.

A 0.02 g PVC sample was soaked in 50 mL of freshly distilled tetrahydrofuran for one week, and then shaken with an ultrasonic extractor for 30 min. After filtration, the supernatant was detected using an UV–Visible spectrometer.

#### 2.5.5. Torque Rheometer Test

According to the ASTM D 2538-79 standard, the impact of MSE-Al on the dynamic rheological property of PVC was investigated by a torque rheometer (RM-200C, Harbin Harp Electric Technology Co., Ltd., Harbin, China). The torque rheometer operating temperature was set to 180 °C, keeping a rotor speed of 35 rpm.

### 2.6. The Capacity for Neutralizing HCl

Conductometric titration experiments were carried out to investigate the capacity of stabilizers to neutralize HCl. The standard solution (6.00 mL of 0.1 mol/L HCl) was diluted with 20.00 mL of ethanol and 10.00 mL of deionized water. MSE-Al, lead salts, and ZnSt_2_ (0.0500 g) were dissolved in this solution with magnetic stirring at 40 °C. The excess HCl solution was back-titrated with 6.00 mL of 0.1 mol/L NaOH standard solution. The conductivity of the solution was measured by a conductivity meter (DDS-307, Shanghai INESA Scientific Instrument Company Limited, Shanghai, China). The volume of NaOH solution corresponding to the minimum conductivity of the solution was the titration endpoint, and the capacity for neutralizing HCl could be calculated by the volume of NaOH solution used.

## 3. Results and Discussion

### 3.1. Characterization of MSE-Al

The FT-IR spectra of MSE and MSE-Al are shown in Figure 1. There are obvious absorption peaks in both spectra at the range of 3200 to 3500 cm^−1^, which correspond to –OH stretching vibration. Two peaks in curve (a) at 1081 and 1019 cm^−1^ belong to –OH bending vibration in MSE. As shown in Figure 1, the peaks at 2918 and 2850 cm^−1^ can be assigned as the C–H stretching vibration, and the peaks at 1467 and 1377 cm^−1^ arise from the bending vibrations of –CH_2_ and –CH_3_, respectively. The peaks at 1740 cm^−1^ in both curve (a) and curve (b) are attributed to C=O stretching vibration of the ester group, indicating that ester groups on MSE and MSE-Al exist. And in curve (b), an intense peak appears at about 1572 cm^−1^ in MSE-Al corresponding to C–O bands, which also exist in aluminum isopropoxide (metal alkoxides) [23]. The peak at 720 cm^−1^ belongs to in-plane rocking vibration of –(CH_2_)_n_–, (n > 4). Khosravi et al. suggested that the C–O–Al bond stretch of aluminum alkoxides appears at 1030–1080cm^−1^ [24]. Curve (b) also shows that the characteristic peak of the C–O–Al bond is superimposed with the two peaks at 1081 and 1019 cm^−1^ as in spectrum (a). The broad peak around 603 cm^−1^ in curve (b) is attributed to the characteristic absorption of the Al–O bond in MSE-Al [23], which confirms the formation of mannitol stearate ester-based aluminum alkoxides (MSE-Al).

### 3.2. Thermal Analysis of MSE-Al

Comparative techniques of thermal analysis were used to evaluate the heat resistance of MSE-Al in a nitrogen atmosphere. TGA, DTG and DTA curves of MSE-Al are shown in Figure 2. There was no apparent weight-loss from 25 to 230 °C in the TGA and DTG curves. This result suggested that MSE-Al had an excellent stability and dispersion at the processing temperature (about 180 °C) of PVC. As shown in Figure 2, the first weight-loss step occurred in the range of 240–340 ℃, and the second weight-loss step occurred in the range of 350–600 °C. After 800 °C, MSE-Al was decomposed into black residue containing zinc oxide and carbon residue with a weight percentage of 12.22%. An endothermic peak could be observed at about 51 °C in this range. In the experiments, when heated to about 50–55 °C, the solid MSE-Al became a paste liquid. Furthermore, the solid MSE-Al could not be crushed with a pulverizer because of the low melting point. Therefore, the first endothermic peak in Figure 2 is likely to be the melting point of MSE-Al. A relatively small peak appeared at around 170 °C, which might correspond to the melting endothermic peak of D-Mannitol (melting point range from 166 to 169 °C). It was due to the mannitol that was not fully reacted in the esterification step (esterification yield is about 96%). Peaks above 250 °C might be the melting point of other compounds produced from the degradation of MSE-Al.

### 3.3. Thermal Stability Tests of MSE-Al on PVC

#### 3.3.1. Appropriate Dosage of MSE-Al

A PVC sample heated at 180 °C will release hydrogen chloride. When the concentration of HCl reaches a certain value, HCl will make the Congo red paper at the top of the PVC sample turn blue. The time when Congo red paper starts to turn blue is defined as the thermal stability time (Δ*T*). In order to determine the optimal dosage of MSE-Al, the stability tests were performed on PVC samples stabilized with different dosages of MSE-Al. As shown in Figure 3, the Δ*T* of pure PVC is only 15 min. With the increase of MSE-Al, the Δ*T* of PVC was obviously prolonged. These results indicated that MSE-Al was able to effectively improve the stability of PVC. However, as the dosage of MSE-Al increased to a certain amount such as 4 phr, the Δ*T* of PVC stabilized by more MSE-Al increased quite slowly. It is well known that there is a low content of unstable structure in PVC molecules. When the dosage of PVC thermal stabilizers reached 4 phr, it was enough to prevent the thermal degradation of PVC. Therefore, increasing the amount of thermal stabilizers would not significantly increase the thermal stability time of PVC. The optimum amount of MSE-Al was determined by the intersection of the two extrapolations. Figure 3 shows that the optimum amount of MSE-Al is 3.22 phr and the resulting Δ*T* is 50.52 min. For comparison with other thermal stabilizers, the dose of MSE-Al was set at 4 phr in this study.

#### 3.3.2. Oven Aging Test of MSE-Al on PVC

The results of the oven thermal aging test of PVC sheets containing different dosages of MSE-Al are shown in Figure 4. It can be seen from Figure 4 that the color of the pure PVC sample starts to turn yellow during the process, and quickly turns to brown and then to black over time with heating at 180 °C. Moreover, the addition of MSE-Al could significantly improve the thermal stability of PVC, and the more the content of MSE-Al, the better the thermal stability of the PVC samples. This implied that MSE-Al could be an excellent long-term thermal stabilizer for PVC. However, the initial color of PVC sheets stabilized with MSE-Al was not very ideal. So, in the following studies, MSE-Al was compounded with ZnSt_2_ to improve the initial whiteness of PVC and to reduce the cost.

### 3.4. Thermal Stability Tests of MSE-Al/ZnSt_2_ on PVC

#### 3.4.1. Congo Red Test

Figure 5 shows the Congo red test results of pure PVC and PVC samples with different ratios of MSE-Al and ZnSt_2_. The Δ*T* of pure PVC was only 15 min. Δ*T* of the PVC sample with addition of 4 phr of MSE-Al increased to 50 min, which indicated that MSE-Al could obviously improve the thermal stability of PVC. The Δ*T* of PVC samples stabilized by 3 phr of MSE-Al and 1 phr of ZnSt_2_ was 58 min, showing an obvious synergistic effect between MSE-Al and ZnSt_2_. F igure 5 also suggests that, with the increase of ZnSt_2_, the thermal stability time of PVC samples obviously decreases, showing that MSE-Al plays a critical role in improving the long-term thermal stability of PVC. As the amount of ZnSt_2_ increased to 4 phr, the Δ*T* was 11 min, even shorter than the thermal stability time of pure PVC, indicating that “zinc burning” might have occurred, which accelerated the degradation of PVC.

#### 3.4.2. Conductivity Test

The HCl produced during the thermal degradation of PVC is carried into the deionized water by nitrogen gas. The conductivity of the deionized water is tested and recorded by a conductivity meter to obtain a curve of the conductivity vs. time. The time from the start of heating to the point where the conductivity begins to change is called the induction time (*T*_i_). The time when the conductivity of deionized water increased to 50 μs/cm is called the thermal stabilization time (*T*_s_) of PVC [17]. Figure 6 shows the conductivity test results of pure PVC and PVC samples with different ratios of MSE-Al and ZnSt_2_. The *T*_i_ and *T*_s_ of PVC samples are also listed in Table 1. As shown in the curve (a) of Figure 6, the *T*_i_ and *T*_s_ of pure PVC samples are 12.7 and 19.9 min, respectively. The *T*_i_ and *T*_s_ of PVC stabilized by 4 phr of MSE-Al increased to 46.3 and 64.3 min, respectively. This meant that MSE-Al might be able to neutralize the HCl produced during the degradation of PVC or inhibit the production of HCl. Curve (c) of Figure 6 is the conductivity curve of the PVC samples stabilized with 3 phr of MSE-Al and 1 phr of ZnSt_2_, which has the longest *T*_i_ and *T*_s_, 49.5 and 84.1 min, respectively. It indicated that there was a strong synergistic effect between MSE-Al and ZnSt_2_. However, with the further increase in dosage of ZnSt_2_, the thermal stability of PVC decreased significantly, and when stabilized with pure 4 phr of ZnSt_2_, the thermal stability of PVC was even worse than pure PVC. This result also proved that the use of pure zinc stearate as a PVC heat stabilizer would catalyze the thermal degradation rate of PVC.

#### 3.4.3. Thermal Aging Test

In the process of PVC thermal degradation, HCl is released and conjugated double bonds are formed simultaneously. With the extension of heating time, the amount of conjugate double bonds is further increased. When the conjugate structure reaches a certain scale, the PVC begins to change color. Generally speaking, the color of PVC changes continuously from white to yellow, brown, and finally black. Figure 7 shows the results of the thermal aging test of PVC sheets containing different mass ratios of MSE-Al and ZnSt_2_ stabilizers. The color of the pure PVC samples started to turn yellow during the process of preparing the PVC sheet with the double roll mill, and quickly turned to brown with heating at 180 °C. As shown in Figure 7, the PVC sheets with 4 phr of MSE-Al added showed a very light pale yellow initial color, would keep the color stable for 110 min, and did not turn black within 360 min, meaning that MSE-Al has an excellent long-term thermal stability on PVC. It can be seen from Figure 7 that the PVC sheets with 3 phr of MSE-Al and 1 phr of ZnSt_2_ show the best initial color and a good long-term stability. The color did not turn black within 110 min, indicating that there was a good synergistic effect between MSE-Al and ZnSt_2_ on the initial color and the long-term thermal stability of PVC. Figure 7 also shows that, if there is ZnSt_2_ in the formula of thermal stabilizers, PVC has good initial whiteness, but the long-term thermal stability worsens with an increase of ZnSt_2_. Furthermore, when the dosage of ZnSt_2_ increased to 3 phr or 4 phr, the PVC sheets began to experience “zinc-burning” phenomenon. The above results indicated that MSE-Al mainly played a role in improving the long-term thermal stability of PVC, while ZnSt_2_ mainly contributed to improving the initial whiteness of PVC. We could assume that, when the complex of MSE-Al and ZnSt_2_ were used as a PVC thermal stabilizer, the reaction rate of MSE-Al, with an active chlorine atom in PVC, and neutralizing hydrogen chloride was not as fast as zinc stearate due to steric hindrance. That is to say, when the dosage of ZnSt_2_ in the complex is low (e.g., 1 phr), due to the low concentration of unstable structures (e.g., allyl chloride) in PVC molecules, the 1 phr of ZnSt_2_ could replace/inactivate the allyl chloride and produce the corresponding amount of ZnCl_2_. Having plenty of hydrogen groups, 3 phr of MSE-Al could form a stable chalet with the low content ZnCl_2_. When the amount of ZnSt_2_ was increased, in addition to replacing allyl chloride, the excess ZnSt_2_ would also react with HCl, which would produce a large amount of ZnCl_2_. The increase in the amount of ZnCl_2_ would make it difficult for the MSE-Al to effectively complex with the ZnCl_2_. The free ZnCl_2_ would shorten the long-term thermal stability of PVC. Therefore, the synergistic effect of 3 phr of MSE-Al and 1 phr of ZnSt_2_ was quite suitable.

#### 3.4.4. UV–Vis Spectroscopy Test

Absorbance of UV–Vis spectra represents the concentration of conjugated double bonds (C_db_), and the position of the absorption peak indicates the length of the conjugated chain. Figure 8 displays the UV–Vis spectra of pure PVC samples and PVC stabilized with different mass ratios of thermal stabilizers heated at 180 °C for 0 or 90 min. Figure 8a shows that the maximum absorption peaks of the six PVC samples are at or near 275 nm, indicating that the dehydrochlorination of PVC produced a conjugated triene structure. Pure PVC had the highest peak height and the C_db_ of pure PVC was the largest, showing that the initial color of pure PVC could be the worst. The smallest was the PVC stabilized with 3 phr of MSE-Al and 1 phr of ZnSt_2_, indicating that PVC stabilized with this component had the best initial color, which was consistent with the results of the thermal aging test.

Figure 8b shows the UV–Vis spectra of PVC samples heated at 180 °C for 90 min. The maximum absorption peak of the six PVC samples shifts to near 300 nm, indicating that PVC samples further decompose to generate a conjugated tetraene structure. Figure 8b also shows that, compared with all other C_db_ of PVC samples, the C_db_ of PVC stabilized with 3 phr of MSE-Al and 1 phr of ZnSt_2_ had the smallest increase. Especially, the C_db_ of pure PVC and PVC stabilized with 4 phr of ZnSt_2_ were more than twice the original. The lowest C_db_ of the PVC sample stabilized with 3 phr of MSE-Al and 1 phr of ZnSt_2_ indicated that the mass ratio (3:1) of MSE-Al and ZnSt_2_ had the best synergistic effect. With the further increase of ZnSt_2_ dosage, the C_db_ of PVC samples gradually increased, indicating that, as the amount of ZnSt_2_ increased, the degree of thermal degradation of PVC increased. It was also found that the C_db_ of PVC samples stabilized by 4 phr of ZnSt_2_ was higher than that of the pure PVC, which indicated that the PVC sample might have undergone the “zinc burning” phenomenon.

#### 3.4.5. Torque Rheometer Test

Figure 9 is the torque rheometer curves of pure PVC and PVC samples stabilized by 4 phr of different thermal stabilizers. In the rheological curve (Figure 9), point A is the filler torque peak, and the greater the torque, the greater the friction of PVC mixture just entering the mixing chamber of the torque rheometer. Point B is the plasticizing torque peak, and its corresponding time indicates the plasticizing effect and difficulty of the PVC mixture. Point C is the balance torque and represents the processing performance (lubricating property) of the PVC mixture. Figure 9 shows that the filler torque values and balance torque values of the PVC samples stabilized by MSE-Al and ZnSt_2_ are much lower than that of pure PVC. It was worth noting that the PVC sample stabilized by 4 phr of ZnSt_2_ had the lowest balance torque, showing that ZnSt_2_ had the best lubricity for PVC. The balance torque of the PVC sample stablized by 4 phr of MSE-Al was much lower than that of pure PVC. All of the results showed that MSE-Al and ZnSt_2_ had good lubrication. This might be due to the presence of a stearic acid radical functional group in MSE-Al and ZnSt_2_, which gave MSE-Al and ZnSt_2_ good lubricity.

Figure 9 also indicates that PVC stabilized by MSE-Al has a plasticizing peak that is 46 s shorter than that of the other PVC samples, showing that MSE-Al has the best excellent plasticizing function. Generally speaking, most of the common plasticizers of PVC are esters, such as phthalate esters [25]. The esters acting as cohesive blocks could increase the compatibility with PVC [26]. For example, Van Oosterhout et al. reported that plasticizers could act as solvents for amorphous regions of PVC, thus the PVC chains in the amorphous regions might become solvated during processing [27]. Therefore, the good plasticizing effect of MSE-Al on PVC was because that MSE-Al had the mannitol stearate ester acting as an efficient plasticizing functional group. Figure 9 shows that the plasticizing time of PVC stabilized by 4 phr of ZnSt_2_ is 255 s, longer than that of pure PVC. Due to the presence of stearate, ZnSt_2_ has an excellent lubricating effect. As for MSE-Al and ZnSt_2_, they have similar polar groups (e.g., –COO^−^), which interacted with the polar fraction of the PVC molecule and decreased polymer-polymer interactions. They should all have good plasticizing properties. However, Figure 9 shows that the plasticizing peaks of PVC stabilized by ZnSt_2_ appears quite later. Perhaps, it was the excellent lubricating property which made the ZnSt_2_ lubricate the PVC blends fully and delay the appearance of plasticizing peaks. In addition, PVC samples stabilized by MSE-Al/ZnSt_2_ (3:1) had excellent plasticizing performance and suitable lubricity, indicating that there was a good synergistic effect between MSE-Al and ZnSt_2_ (3:1) on plasticizing and lubricity.

### 3.5. Thermal Stabilizing Mechanism of MSE-Al

Metal alkoxides are strong alkalis and good nucleophiles. The ability of metal alkoxides to neutralize HCl is strong, but they are not suitable to be used as PVC thermal stabilizers due to their easy hydrolysis [28]. However, MSE-Al is a polyol ester-based metal alkoxide, which cannot be easily hydrolyzed due to the property of polyhydric hydroxyl metal alkoxides and the introduction of a stearate, and it can then act as a thermal stabilizer for PVC.

#### 3.5.1. Neutralize HCl

One of the main functions of MSE-Al as a PVC thermal stabilizer is to neutralize HCl. In order to investigate the ability of MSE-Al to absorb HCl, a conductometric titration experiment [29] was used to compare it with lead salts and ZnSt_2_. As shown in Table 2, the capacity of MSE-Al to absorb HCl is distinctly lower than that of lead salts and ZnSt_2_, indicating that the main role of MSE-Al to protect PVC may be to react with the unstable chlorine atom in the PVC chain to prevent the degradation of PVC, just as the other polyol-based metal alkoxides do [30].

Figure 7 shows that the “zinc-burning” phenomenon is not observed in PVC stabilized by pure MSE-Al. The possible reason was that no free AlCl_3_ was produced in the degradation of PVC stabilized by MSE-Al. Based on the results it was deduced that MSE-Al as a PVC thermal stabilizer could be used for neutralizing HCl through the equation in Scheme 3. In our previous work [30], we studied the mechanism of the reaction between metal alkoxides and HCl with the help of quantum chemical calculations. The metal atoms (e.g., aluminum atom) of metal alkoxides had quite low electron density, and the chlorine atom of HCl had high electron density. Thus, the aluminum atoms in MSE-Al had a considerable tendency to undergo an electrophilic reaction with the chlorine atom of HCl. At the same time, the alkyl oxygen (having high electron density) of MSE-Al would attack the hydrogen atom (having low electron density) of HCl. Finally, the bond of H–Cl was broken to form a new chemical bond of –O–H and –Al–Cl. Due to its good complexing ability, the hydroxyl group will form chelate with –Al–Cl, as shown in Scheme 3. One mole of MSE-Al could, theoretically, neutralize three moles of HCl.

#### 3.5.2. Replacement of the Labile Chlorine Atoms in the PVC

Due to the presence of high electronegative alkoxy groups, MSE-Al had a tendency to undergo nucleophilic reactivity, and could replace the labile chlorine atoms in the PVC [30]. Figure 10 shows the FT-IR spectra of pure PVC and PVC samples stabilized by MSE-Al. After having been heated at 180 °C for 30 min, both the pure PVC and PVC sample stabilized by MSE-Al had a stretching vibration peak of C=C around 1672 cm^−1^, suggesting that PVC degrades to form a conjugated double bond. Compared with the pure PVC sample, a new peak appeared at 1162.5 cm^−1^ for the PVC sample stabilized with MSE-Al, which corresponded to the stretching vibration of C–O–C [31]. These results showed that MSE-Al replaced the labile chlorine atoms in the PVC chains. The reaction between MSE-Al and labile chlorine atoms can be described by the equation in Scheme 4.

#### 3.5.3. Formation of a Complex with AlCl_3_ and ZnCl_2_

In order to verify that MSE-Al could form a stable complex with AlCl_3_ and ZnCl_2_, the thermal aging test of PVC samples stabilized by pure AlCl_3_, ZnCl_2_, and AlCl_3_/MSE-Al, were tested, and the results are shown in Figure 11. The color of PVC stabilized with pure AlCl_3_ was light gray at the beginning, and gradually changed into light brown, and dark brown at last. The results showed that although the AlCl_3_ had no thermal stabilization effect on PVC, it did not cause the “zinc-burning” phenomenon, indicating that the existence of AlCl_3_ did not catalyze the dehydrogenation of PVC. It was the reason why the PVC stabilized with MSE-Al had an excellent long-term stability effect. On the contrary, the color of PVC samples stabilized by pure ZnCl_2_ turned into absolute black after being heated for 10 min, showing the occurrence of the “zinc-burning” phenomenon. Figure 11 also shows that the combination of AlCl_3_ and MSE-Al can inhibit PVC from turning black quickly, indicating that the complexes can form between them. The PVC samples stabilized by ZnCl_2_/MSE-Al with a mass ratio of 3:1 were light gray white in the beginning, and turned into absolute black after being heated for 30 min. Furthermore, the color of PVC stabilized by ZnCl_2_/MSE-Al with a mass ratio of 1:3 did not turn into black within 90 min, suggesting that the hydroxyl of MSE-Al could form a complex with ZnCl_2_, thus suppressing the occurrence of “zinc burning”. The possible mechanism can be represented by Scheme 5. This fact also explains why there was a good synergy between MSE-Al and ZnSt_2_.

## 4. Conclusions

MSE-Al was synthesized through an alcohol exchange reaction, and investigated by means of a number of characterization methods. The introduction of a stearate radical reduced the melting point of the metal alkoxides, increased the dispersibility in PVC, and improved the lubricity between PVC powders. Furthermore, the torque rheometer test results demonstrated that MSE-Al was able to significantly enhance the plasticizing effect and lubricity of PVC.

The thermal stability of PVC samples stabilized by MSE-Al, and MSE-Al/ZnSt_2_ was also evaluated by Congo red tests, conductivity measurements, thermal aging tests, and UV-vis spectroscopy tests. The results showed that MSE-Al could significantly improve the long-term thermal stability of PVC, attributed to the fact that MSE-Al replaced the labile chlorine atoms in the PVC chain and had an ability to neutralize HCl. A synergistic effect between MSE-Al and ZnSt_2_ was observed. It was also found that PVC stabilized by 3 phr of MSE-Al and 1 phr of ZnSt_2_ showed a good initial color and excellent long-term thermal stability. This was because ZnSt_2_ had a strong ability to absorb hydrogen chloride, and MSE-Al would not only replace unstable chlorine atoms in the PVC chain, but also chelate the ZnCl_2_ produced by absorption of HCl by ZnSt_2_, thereby inhibiting the “zinc burning” phenomenon.

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
