# Peer review of "Design and Synthesis of a New Mannitol Stearate Ester-Based Aluminum Alkoxide as a Novel Tri-Functional Additive for Poly(Vinyl Chloride) and Its Synergistic Effect with Zinc Stearate"

_polymers, 2019, doi:10.3390/polym11061031_

Reviewer 1 Report

In this manuscript titled “Design and synthesis of a new mannitol stearate ester-based aluminum alkoxides as a novel tri-functional additive for poly(vinyl chloride) and its synergistic effect with zinc stearate”, the author synthesized a new mannitol stearate ester‐based aluminum alkoxides (MSE‐Al) as a novel additive for PVC. MSE‐Al could be used as three crucial additives including thermal stabilizer, plasticizer, and lubricant for processing poly (vinyl chloride) (PVC). Meanwhile, the use of MSE-Al with low dosage of ZnSt2could improve the initial whiteness of PVC, which solved the problem that the initial color of PVC sheets stabilized with only MSE‐Al was not very good. This manuscript is original. However, a few questions are list below.

1. In Line 182, Page 5, I think it would be better to mark the peaks at 1467 and 1377 cm-1 in Figure 1.

2. In Lines 214 and 215, Page 7, “However, as the dosage of MSE-Al increased to a certain amount, the ΔT of PVC stabilized with more MSE‐Al almost would not increase.” Why? I think the author should explain this phenomenon carefully.

3. In Lines 336-338, Page 12, “The plasticizing time of PVC stabilized by 4 phr of ZnSt2was 255 s, longer than that of pure PVC, indicating that ZnSt2has no plasticizing effect on PVC.” Since the plasticizing time of PVC stabilized by 4 phr of ZnSt2is longer than that of pure PVC, I don’t think ZnSt2has no plasticizing effect on PVC. I think the author should reexplain the effect of ZnSton the plasticization of PVC. And why.

4. In Page 13, I think it would be better to explain the process of neutralizing HCl with MSE-Al in more detail according to Scheme 3.

5. In Lines 263-265, Page 9, “This result also proved that the use of pure zinc stearate as a PVC heat stabilizer would catalyze the thermal degradation rate of PVC.” Then, why did the PVC sheets with 3 phr of MSE-Al and 1 phr of ZnSt2show the best initial color and a good long-term stability? I suggest that the author explain the thermal stabilizing mechanism of MSE-Al with low dosage of ZnSt2.

6. In Lines 28 and 29, Page 1, the first letter of each key word should be capitalized.

7. In Lines 109 and 118, Page 3, I think it would be better to provide the corresponding names below the materials and provide the reaction conditions (such as reaction time and temperature) above the arrows in Scheme 1 and Scheme 2.

8. In Figure 4 (Line 220, Page 7), I suggest that the author adjust the pictures in “4 phr”line to keep it in the same form as other pictures.

9. In Page 9, I think the font size of the XY coordinate names is too big in Figure 6.

10. In Page 12, I think it would be better to keep the color of “255 s” the same as the corresponding line color in Figure 9.

Author Response

Reply to Reviewer#1s Comments

In this manuscript titled “Design and synthesis of a new mannitol stearate ester-based aluminum alkoxides as a novel tri-functional additive for poly(vinyl chloride) and its synergistic effect with zinc stearate”, the author synthesized a new mannitol stearate ester‐based aluminum alkoxides (MSE‐Al) as a novel additive for PVC. MSE‐Al could be used as three crucial additives including thermal stabilizer, plasticizer, and lubricant for processing poly (vinyl chloride) (PVC). Meanwhile, the use of MSE-Al with low dosage of ZnSt2 could improve the initial whiteness of PVC, which solved the problem that the initial color of PVC sheets stabilized with only MSE‐Al was not very good. This manuscript is original. However, a few questions are list below.

1.In Line 182, Page 5, I think it would be better to mark the peaks at 1467 and 1377 cm-1 in Figure 1.

Reply: 

We are very grateful to the Reviewer#1 for pointing out this shortcoming. We have marked the peaks at 1467 and 1377 accordingly. We have also enlarge the front size of the letter in Figure 1 to make it easier to read.

Please refer to the revised Figure 1 in Page 6 of the revised manuscript.

2.In Lines 214 and 215, Page 7, “However, as the dosage of MSE-Al increased to a certain amount, the ΔT of PVC stabilized with more MSE-Al almost would not increase.” Why? I think the author should explain this phenomenon carefully.

Reply:

We thank the reviewer for the good question and we are sorry for our deficient expression. We have change “almost would not increase” into “increased quite slowly”, and added the following explanation.

It is well known that there is low content of unstable structure in PVC molecules. When the dosage of PVC thermal stabilizers reached 4 phr, it was enough to prevent the thermal degradation of PVC. Therefore, increasing the amount of thermal stabilizers would not significantly increase the thermal stability time of PVC.

Please refer to the revised part in Page 7 (line 246-249) of the revised manuscript.

3. In Lines 336-338, Page 12, “The plasticizing time of PVC stabilized by 4 phr of ZnSt2 was 255 s, longer than that of pure PVC, indicating that ZnSt2 has no plasticizing effect on PVC.” Since the plasticizing time of PVC stabilized by 4 phr of ZnSt2 is longer than that of pure PVC, I don’t think ZnSt2 has no plasticizing effect on PVC. I think the author should reexplain the effect of ZnSt2 on the plasticization of PVC. And why.

Reply:

We appreciate Reviewer’s insightful comments. We have delete the sentence indicating that ZnSt2 has no plasticizing effect on PVC and added the following explanation.

Due to the presence of stearate, ZnSt2 has excellent lubricating effect. As for MSE-Al and ZnSt2, they had the similar polar groups (e.g., -COO-) which interacted with the polar fraction of PVC molecule and decreased polymer-polymer interactions. They should all have good plasticizing properties. However, Figure 9 shows that the plasticizing peaks of PVC stabilized by ZnSt2 appears quite later. Perhaps, it was the excellent lubricating property which made the ZnSt2 lubricate the PVC blends fully and delay the appearance of plasticizing peaks.

Please refer to the revised part in Page 12 (line 387-392) of the revised manuscript.

4. In Page 13, I think it would be better to explain the process of neutralizing HCl with MSE-Al in more detail according to Scheme 3.

Reply:

We are very grateful to the Reviewer for the good suggest. We have added the following explanation.

The Figure 7 shows that “zinc-burning” phenomenon is not observed in PVC stabilized by pure MSE-Al. The possible reason was that no free AlCl3 produced in the degradation of PVC stabilized by MSE-Al. Based on the results it was deduced that MSE-Al as a PVC thermal stabilizer could be used for neutralizing HCl through the equation in Scheme 3. In our previous work [30], we had study the mechanism of the reaction between metal alkoxides and HCl with the help of quantum chemical calculations. The metal atoms (e.g. aluminum atom) of metal alkoxides had quite low electron density, and the chlorine atom of HCl had high electron density. Thus, the aluminum atoms in MSE-Al had a considerable tendency to undergo an electrophilic reaction with the chlorine atom of HCl. At the same time, the alkyl oxygen (having high electron density) of MSE-Al would attack the hydrogen atom (having low electron density) of HCl. Finally, the bond of H-Cl was broken to form new chemical bond of -O-H and -Al-Cl. Due to having good complexing ability, the hydroxyl group will form chelate with -Al-Cl just as shown in Scheme 3. One mole of MSE-Al could theoretically neutralize three moles of HCl.

Please refer to the revised part in Page 13 (line 410-422) of the revised manuscript.

5. In Lines 263-265, Page 9, “This result also proved that the use of pure zinc stearate as a PVC heat stabilizer would catalyze the thermal degradation rate of PVC.” Then, why did the PVC sheets with 3 phr of MSE-Al and 1 phr of ZnSt2 show the best initial color and a good long-term stability? I suggest that the author explain the thermal stabilizing mechanism of MSE-Al with low dosage of ZnSt2.

Reply:

We are very appreciated for the Reviewers good suggest. We have added the following explanation.

We could assume that, when the complex of MSE-Al and ZnSt2 were used as PVC thermal stabilizer, the reaction rate of MSE-Al with active chlorine atom in PVC and neutralizing hydrogen chloride was not as fast as zinc stearate due to steric hindrance. That is to say, when the dosage of ZnSt2 in the complex is low (e.g. 1 phr), due to the low concentration of unstable structures (e.g. allyl chloride) in PVC molecules, the 1 phr of ZnSt2 could replace/inactivate the allyl chloride and produce the corresponding amount of ZnCl2. Having plenty of hydrogen group, 3 phr of MSE-Al could form stable chalet with the low content ZnCl2. When the amount of ZnSt2 was increased, in addition to replacing allyl chloride, the excess ZnSt2 would also react with HCl, which would produce a large amount of ZnCl2. The increase of the amount of ZnCl2 would make it difficult for the MSE-Al to effectively complex with the ZnCl2. The free ZnCl2 would shorten the long-term thermal stability of PVC. Therefore, the synergistic effect of 3 phr of MSE-Al and 1 phr of ZnSt2 was quite suitable.

Please refer to the revised part in Page 10 (line 323-334) of the revised manuscript.

6. In Lines 28 and 29, Page 1, the first letter of each key word should be capitalized.

Reply:

We thank Reviewer for pointing out the problem. We have revised this accordingly.

Please refer to the revised Key words in Page 1 of the revised manuscript.

7. In Lines 109 and 118, Page 3, I think it would be better to provide the corresponding names below the materials and provide the reaction conditions (such as reaction time and temperature) above the arrows in Scheme 1 and Scheme 2.

Reply:

We are very appreciated for the Reviewer’s good suggest. We have added the corresponding name and provided the reaction conditions.

Please refer to the revised Scheme 1 and Scheme 2 in Page 3 and Page 4, respectively.

8. In Figure 4 (Line 220, Page 7), I suggest that the author adjust the pictures in “4 phr”line to keep it in the same form as other pictures.

Reply:

We thank Reviewer for the very meticulous observation and for pointing out this error. We have corrected this accordingly.

Please refer to the revised Figure 4 in Page 8 of the revised manuscript.

9. In Page 9, I think the font size of the XY coordinate names is too big in Figure 6.

Reply:

We are very appreciated for the Reviewer’s good suggest. We have amended this accordingly.

Please refer to the revised Figure 6 in Page 9 of the revised manuscript.

10. In Page 12, I think it would be better to keep the color of “255 s” the same as the corresponding line color in Figure 9.

Reply:

We are very appreciated for the Reviewer’s good suggest. We have corrected this accordingly.

Please refer to the revised Figure 9 in Page 12 of the revised manuscript.

We appreciate for Reviewer's work earnestly, and hope that the corrections will meet with approval. Once again, thank you very much for Reviewer's good comments and suggestions.

Reviewer 2 Report

The research work done by Degang Li and Hongqi Sun co-workers “Design and synthesis of a new mannitol stearate ester-based aluminum alkoxides as a novel tri functional additive for poly(vinyl chloride) and its synergistic effect with zinc stearate” is a good research work to study and designed for additive for poly(vinyl chloride). Authors documented a new process and the synthesis of mannitol stearate ester-based aluminum alkoxides. The major advantage of the present documented report is the synthesized material can be used as novel tri functional additive for poly(vinyl chloride). There are not many literature procedures, similar to present documented report in the literature for new developments as additives. Given the importance of practicality for this work, I recommend the publication of this manuscript in the Polymers.

Author Response

Reply to Reviewer#2s Comments

The research work done by Degang Li and Hongqi Sun co-workers “Design and synthesis of a new mannitol stearate ester-based aluminum alkoxides as a novel trifunctional additive for poly(vinyl chloride) and its synergistic effect with zinc stearate” is a good research work to study and designed for additive for poly(vinyl chloride). Authors documented a new process and the synthesis of mannitol stearate ester-based aluminum alkoxides. The major advantage of the present documented report is the synthesized material can be used as novel tri functional additive for poly(vinyl chloride). There are not many literature procedures, similar to present documented report in the literature for new developments as additives. Given the importance of practicality for this work, I recommend the publication of this manuscript in the Polymers.

Reply:

We thank Reviewer for his/her very positive comments and significant contribution to our manuscript.

Reviewer 3 Report

In this manuscript, a new mannitol stearate ester‐based aluminum alkoxides (MSE‐Al) was designed and synthesized as a novel additive for PVC. The results show that MSE‐Al has good synergetic effect with ZnSt2 on the thermal stability of PVC. The thermal stabilization mechanism and synergism effect of MSE‐Al with ZnSt2 were also discussed. The work is significant and novel. But in my opinion, the following comments should be considered before the manuscript is accepted.

Comment 1:

Whether unreacted raw materials are removed or not? And the productivity in synthesize should also be mentioned.

Comment 2:

The title below the figure is Figure, which is Fig. in the discussion, the two should be unified.

Comment 3:

In line 181,” the peaks at 2920 and 2851 cm-1”, which is not consistent with the Figure 1. The peaks at 1467 and 1377 cm-1 were not marked in Figure 1.

Comment 4:

DTA curves in Figure 2 had many small peaks. It is controversial that the melting point of MSE-Al is about 51 oC

Comment 5:

In line 215,”the ΔT of PVC stabilized with more MSE‐Al almost would not increase” is not true as shown in Figure 3. No platform existed in Figure 3. It just increased slowly when the dosage of Al-MSE was large.

Comment 6:

Separation between headings and units in graphs or tables should be unified, using brackets or slashes.

Comment 7:

The abscissa range in Figure 8 is not consistent with the description in Part 2.5.4.

Comment 8:

In line 303 and 304,” Fig. 8b also shows that, except that the Cdb of PVC stabilized with 3 phr of MSE‐Al and 1 phr of ZnSt2 was remained unchanged” didn’t correctly described, compared with Figure 8, which also slightly increased, and the shape of the curve was not the same before and after aging.

Comment 9:

The low balance torque of PVC sample with 4 phr of ZnSt2 and MSE‐Al may cause by the strong degradation and low molecular weight of PVC chains. Additional evidence is required to prove the lubricity effect.

Comment 10:

Further English Improvement. Academic English revision is needed for accurate description and report.

Author Response

Reply to Reviewer#3s Comments

In this manuscript, a new mannitol stearate ester‐based aluminum alkoxides (MSE-Al) was designed and synthesized as a novel additive for PVC.The results show that MSE‐Al has good synergetic effect with ZnSt2 on the thermal stability of PVC. The thermal stabilization mechanism and synergism effect of MSE‐Al with ZnSt2 were also discussed. The work is significant and novel. But in my opinion, the following comments should be considered before the manuscript is accepted.

Comment 1:

Whether unreacted raw materials are removed or not? And the productivity in synthesize should also be mentioned.

Reply:

We are very appreciated for the Reviewer’s good suggest. The yield of mannitol stearate ester (MSE) is about 96±1 %. The unreacted trace raw materials are stearic acid and mannitol. In industrial applications of PVC thermal stabilizers, stearic acid and polyol are the commonly used lubricant and complexing agents, respectively. Therefore, residual trace stearic acid and mannitol have no bad effect on the thermal stability of PVC, so we did not remove them. Furthermore, we had used a variety of methods and solvents to try to separate the unreacted raw materials, the results showed that the products could not be dissolved in numerous different solvents. Since the trace unreacted raw materials do not adversely affect the thermal stability of PVC, in order to reduce the production costs, we no longer purify the product.

To determine the yield of the esterification reaction for MSE, the volume of water produced after the completion of the esterification reaction is measured. By comparing the actual water volume (Va) and the theoretical water volume (Vt), it can calculate the yield of MSE using Equation (1). The yield of MSE from Equation (1) was 96±1 %.

                                                               (1)

Since aluminum isopropoxide and the produced isopropanol were soluble in absolute ethanol, while MSE-Al was insoluble in absolute ethanol at room temperature, the crude product of MSE-Al was smashed and put into absolute ethanol. The mixture was filtered after stirring for 1 h. The filtrate components were ethanol, isopropanol and unreacted aluminum isopropoxide. After evaporated the liquid at 90 oC, the residual solid (aluminum isopropoxide) was measured by inductively coupled plasma mass spectrometer (ICP-MS). By comparing the molar amount of aluminum in the unreacted aluminum isopropoxide (Mu/mmol) and the total molar amount of aluminum (10 mmol) in the reactant, the yield of MSE-Al could be calculated through Equation (2). The yield of MSE-Al from Equation (2) was 92±2 %.

                                                              (2)

We have added the corresponding content. Please refer to the revised part in Page 3 and 4 (line 110-115, 126-136) of the revised manuscript.

Comment 2:

The title below the figure is Figure, which is Fig. in the discussion, the two should be unified.

Reply:

We thank Reviewer for pointing this error. We have amended this accordingly.

Comment 3:

In line 181,” the peaks at 2920 and 2851 cm-1”, which is not consistent with the Figure 1. The peaks at 1467 and 1377 cm-1 were not marked in Figure 1.

Reply:

We thank Reviewer for his/her very meticulous observation and for pointing out this error. We have corrected this accordingly and marked the 1467 and 1377 cm-1 in Figure 1.

Please refer to the revised Figure 1 in Page 6 of the revised manuscript.

Comment 4:

DTA curves in Figure 2 had many small peaks. It is controversial that the melting point of MSE-Al is about 51oC

Reply:

We thank Reviewer for his/her good comments. In the experiments, when heated to about 50-55 oC, the solid MSE-Al became a paste liquid. Furthermore, the solid MSE-Al couldn’t be crushed with a pulverizer because of the low melting point. Therefore, the first endothermic peak in Figure 2 is likely to be the melting point of MSE-Al. A quite small peak appeared at around 170 oC, which might correspond to the melting endothermic peak of D-Mannitol (Melting point range from 166 to 169 oC). It was due to the mannitol that was not fully reacted in the esterification step (esterification yield is about 96%). Peaks above 250 oC might be the melting point of other compounds produced from the degradation of MSE-Al.

We have added the corresponding discuss. Please refer to the revised part in Page 7 (line 226-233) of the revised manuscript.

Comment 5:

In line 215,”the ΔT of PVC stabilized with more MSE‐Al almost would not increase” is not true as shown in Figure 3. No platform existed in Figure 3. It just increased slowly when the dosage of Al-MSE was large.

Reply:

We thank the reviewer for the good question and we are sorry for our deficient expression. We have change “almost would not increase” into “increased quite slowly”, and added the following explanation.

It is well known that there is low content of unstable structure in PVC molecules. When the dosage of PVC thermal stabilizers reached 4 phr, it was enough to prevent the thermal degradation of PVC. Therefore, increasing the amount of thermal stabilizers would not significantly increase the thermal stability time of PVC.

Please refer to the revised part in Page 8 (line 246-249) of the revised manuscript.

Comment 6:

Separation between headings and units in graphs or tables should be unified, using brackets or slashes.

Reply:

We thank the reviewer for the good question. We have unified the separation and used the slashes in all of the in graphs and tables.

Comment 7:

The abscissa range in Figure 8 is not consistent with the description in Part 2.5.4.

Reply:

We thank Reviewer for his/her very meticulous observation and for pointing out this error. We have corrected the abscissa range in Figure 8.

Please refer to the revised in Part 2.5.4 (line 182) of the revised manuscript. The abscissa range is used in the range of 270-400 nm uniformly.

Comment 8:

In line 303 and 304,” Fig. 8b also shows that, except that the Cdb of PVC stabilized with 3 phr of MSE-Al and 1 phr of ZnSt2 was remained unchanged” didn’t correctly described, compared with Figure 8, which also slightly increased, and the shape of the curve was not the same before and after aging.

Reply:

We thank the reviewer for the good question and we are sorry for our inaccurate expression. We have revised them carefully. 

We have changed “Fig. 8b also shows that, except that the Cdb of PVC stabilized with 3 phr of MSE-Al and 1 phr of ZnSt2 was remained unchanged, all the other Cdb of PVC samples changed greatly. into “Figure 8b also shows that, compared with all other Cdb of PVC samples, the Cdb of PVC stabilized with 3 phr of MSE-Al and 1 phr of ZnSt2 had the smallest increase”. Please refer to line 351-353 in the revised manuscript.

Furthermore, we have described the change of the shape in Line 349-351.

Comment 9:

The low balance torque of PVC sample with 4 phr of ZnSt2 and MSE-Al may cause by the strong degradation and low molecular weight of PVC chains. Additional evidence is required to prove the lubricity effect.

Reply: 

We thank the reviewer for the good question. Because having long alkyl chain, organic compounds containing stearate generally have good lubricity, for example, stearic acid, zinc stearate and calcium stearate. In the industrial production of PVC thermal stabilizers, stearic acid and zinc stearate are commonly used lubricants.

We found in the torque rheometer experiment that the color of PVC didn’t change after the test. In addition, if ZnSt2 and MSE-Al made PVC degrade, the curves on the far right of Figure 9 would rise significantly (the balance torque increases rapidly). However, the balance torque of curves didn’t change obviously at 400 s, indicating that PVC didn’t degrade.

In summary, ZnSt2 is the traditional lubricant. As for the MSE-Al, the presence of stearate makes MSE-Al have good lubrication.

Comment 10:

Further English Improvement. Academic English revision is needed for accurate description and report.

Reply:

We apologized for our poor English. According to the reviewer's good instruction, we have sought help from a person proficient in the English language and amended the manuscript carefully. We hope that the interpretation of results now is acceptable for the next review process. 

We appreciate for Reviewer's work earnestly, and hope that the corrections will meet with approval. Once again, thank you very much for Reviewer's good comments and suggestions.

Round  2

Reviewer 1 Report

I think the author has carefully revised the manuscript and the manuscript can be published in Polymers.

Regards and best wishes,

Reviewer 3 Report

This manuscript is suitable for publication in its present form.